# Leber Hereditary Optic Neuropathy: Molecular Pathophysiology and Updates on Gene Therapy

**DOI:** 10.3390/biomedicines10081930

**Published:** 2022-08-09

**Authors:** Sheng-Chu Chi, Hui-Chen Cheng, An-Guor Wang

**Affiliations:** 1Department of Ophthalmology, Taipei Veterans General Hospital, Taipei 112, Taiwan; 2Department of Ophthalmology, School of Medicine, National Yang Ming Chiao Tung University, Taipei 112, Taiwan; 3Program in Molecular Medicine, College of Life Sciences, National Yang Ming Chiao Tung University, Taipei 112, Taiwan; 4Department of Life Sciences and Institute of Genome Sciences, College of Life Sciences, National Yang Ming Chiao Tung University, Taipei 112, Taiwan; 5Brain Research Center, National Yang Ming Chiao Tung University, Taipei 112, Taiwan

**Keywords:** leber hereditary optic neuropathy, molecular pathophysiology, gene therapy

## Abstract

Molecular pathophysiology of LHON was reviewed and the current status of gene therapy for LHON is updated.

## 1. Introduction

Leber hereditary optic neuropathy (LHON) is a maternally transmitted genetic disorder caused by mutation of mitochondrial DNA (mtDNA). This vision-threatening disease typically presents in male patients between 15–35 years old and causes subacute central vision loss [1,2]. In over 90% of cases, LHON is caused by one of three mitochondrial primary mutations: m.3460G > A (MTND1), m.11778G > A (MTND4), and m.14484T > C (MTND6) [3,4]. The mutation compromises the respiratory chain in mitochondria, which leads to retinal ganglion cell loss and optic neuropathy [5]. Among these mutations, the m.11778G > A mutation accounts for about 75% of LHON in Western countries and is associated with poor visual outcomes, with a spontaneous visual recovery rate ranging from 4 to 23% in previous studies [4,5,6,7]. In most cases of the m.11778G > A mutation, vision typically deteriorates to worse than 20/200 [8].

Human mitochondrial complex I (NADH dehydrogenase; ND), composed of 45 subunits, is the first and largest oxidative phosphorylation complex [9]. With energy released by transferring electrons from complex I to complex III/IV, protons are pumped across the inner membrane of mitochondria, and an electrochemical gradient is created, leading to adenosine triphosphate (ATP) synthesis with the help of H+-translocating ATP synthase (complex V) [10]. Most LHON patients have a single mitochondrial DNA (mtDNA) mutation, leading to complex I dysfunction in the electron transport chain [11]. The common primary mtDNA mutations 3460, 11,778, and 14,484, affecting ND1, ND4, and ND6 subunits of complex I, may cause respiratory dysfunction of the electron transport chain (Figure 1). To assess the cellular function of these mutants, cybrids have been developed for in vitro studies. Transmitochondrial technology has been used to establish cybrid cell lines by repopulating exogenous mitochondria in the Rho^0^ cell line, which is depleted of its own mtDNA using ethidium bromide [12]. Cybrid cell lines have been widely used to study the cellular biochemical effects of mtDNA mutations in vitro since they have the same nuclear background with different mtDNA mutations. One study [13], using biochemical analysis, showed a 79% reduction in complex I activity in the 3460 mutant cybrid, a 20% reduction in the 11,778 mutant cybrid, and no obvious effect in the 14,484 mutant cybrid. However, other studies did not show a significant reduction in cybrids with the 11,778 mutation [14,15]. Severe impairment of complex I-driven ATP synthesis has been noted in cybrids with these three pathogenic mutations, but they might be effectively compensated by glycolysis and complex II/glycerol 3-phosphate dehydrogenase pathways, with the total cellular ATP content not significantly decreased [16,17]. In addition, reactive oxygen species (ROS) production is increased in cybrids carrying these three primary mutations, with complex I and complex III as the main sources of superoxide production [18,19]. With compromised complex I activity and reduced ATP synthesis, LHON cybrids showed increased sensitivity to apoptotic cell death mediated by fatty acid synthesis (FAS) or apoptosis-inducing factor (AIF) [20,21]. Though prior studies investigating cybrid cell lines clarified many molecular biochemical mechanisms of LHON, some crucial questions remain unsolved. Several transgenic mice models of mitochondrial disease were produced to address these questions [22,23,24,25,26]. In 2012, Lin et al., successfully produced a mutant ND6 P25L (G13997A) transgenic mouse model by introducing an mtDNA mutation into the germ line of female mice [26]. These transgenic mouse models may reveal the molecular mechanism of pathogenesis and could be used to test potential candidate therapies.

As for the current treatment of LHON, idebenone (Raxone, Santhera GmbH) has been prescribed for patients in Europe. Idebenone can transfer electrons directly to mitochondrial complex III. Thus, it replaces the dysfunctional mitochondrial complex I and restores the function of energy production in mitochondria [27]. A clinical trial of idebenone showed that the secondary endpoint of change in best-corrected visual acuity was significantly better with Raxone use, while the primary endpoint of best recovery in visual acuity did not reach statistical significance [28].

In recent years, gene therapy has been regarded as a potential solution for LHON. However, gene therapy for mitochondrial DNA mutations has been a challenging issue since the presence of a double membrane may prevent or decrease the delivery of genetic materials into mitochondria [29]. In addition, a single somatic cell may contain hundreds of mitochondria with a multicopy genome, which makes the feasibility of direct mitochondrial delivery of nucleic acids questionable. To circumvent this difficulty, allotopic expression was adopted by Dr. Guy’s group in Miami in 2002 [30,31,32]. This technology uses a transgene of nuclear DNA (nDNA) encoding the wild-type mitochondrial *MTND4* protein. With the aid of an adeno-associated virus (AAV) vector, the transgene is introduced into the nucleus of retinal ganglion cells and transcribes messenger RNA. With a mitochondrial targeting sequence (MTS), the messenger RNA is then shuttled from the nucleus to the outer membrane of the mitochondria, where the wild-type *MTND4* protein is translated and incorporated into the mitochondrial respiratory chain complex [33] (Figure 2). In cybrids harboring the G11778A mutation, AAV-ND4 helps to increase ATP synthesis and improves the survival rate three-fold [33]. Bonnet C et al., reported that allotopic expression helped to restore respiratory complex I/V activity, ATP synthesis, and the cell survival rate in cultured skin fibroblasts isolated from two LHON patients harboring mutations in ND1 (3460) or ND4 (11,778) and one NARP (neurogenic muscle weakness, ataxia and retinitis pigmentosa) patient with the T8993G *ATP6* mutation [30,31]. Later, Guy J et al., developed an animal model of LHON by injecting rAAV-mutant ND4 (G11778A, R340H) into the vitreous cavities of adult wild-type mice in 2007. They found that mutant ND4 initially disrupts the mitochondrial cytoarchitecture, increases reactive oxygen species, and induces optic nerve head swelling, ending with apoptosis of retinal ganglion cells and optic atrophy [34]. Ellouze S et al., also created an animal model of LHON in 2008 by introducing a mutant ND4 gene (G11778A) into adult rat eyes by in vivo electroporation, which caused a 40% decrease in RGCs [35]. Guy J et al., tested the efficiency and rapidity of AAV-ND4 vectors in adult mice in 2010 with either self-complementary adeno-associated virus 2 (scAAV2) capsids or single-stranded AAV2 (ssAAV2) capsids. Constructs were injected into the vitreous cavities of adult wild-type mice, and they found typical perinuclear mitochondrial expression of scAAV-ND4 in 91% of Thy1.2-positive RGCs, while ssAAV-ND4 was expressed in 51% [32]. Then, Guy J et al., attempted to rescue vision loss in a mutant LHON mice model in 2014 by intravitreal injection of scAAV2-P1ND4v2. They found that there was a significant rescue of retinal function and prevention of the demise of RGCs and optic axons with the use of scAAV2-P1ND4v2. With the rat LHON model, Cwerman-Thibault H et al., delivered a recombinant adeno-associated viral vector containing human ND4 (rAAV2/2-*ND4*) in 2015 and demonstrated that human ND4 protein can be imported inside the mitochondria and did not lead to harmful effects in the rat eye. In addition, human ND4 protein can be assembled into complex I, preserves its function, and prevents retinal ganglion cell degeneration [36]. Subsequently, several groups conducted clinical trials based on the concept and technology of allotopic expression, including Wuhan in China, Miami in the USA, and Paris in France.

To date, the most abundant clinical data regarding gene therapy for LHON are from a group from GenSight Biologics in Paris, France. GenSight Biologics was established in 2012 in Paris, France. They initiated a phase 1/2 dose-escalation cohort study (NCT02064569) in 2014 and proved the safety of rAAV2/2-ND4 (GS010) [37,38]. They further launched two phase 3 studies, RESCUE (NCT02652767) and REVERSE (NCT02652780), in 2016 as randomized, double-masked, sham-controlled studies with a single intravitreal injection of rAAV2/2-ND4 (GS010, LUMEVOQ). Long-term follow-up of LHON cohorts in RESCUE/REVERSE trials was registered as the RESTORE trial (NCT03406104) [31]. GenSight Biologics launched another new multicenter, double-masked, randomized controlled phase 3 trial, REFLECT (NCT03293524), in the United States, Europe, and Taiwan in 2018 [39]. 

The Miami group (Bascom Palmer Eye Institute, University of Miami) conducted a phase 1 clinical trial (NCT02161380) in 2014. This was an open-label dose-escalation study with scAAV2-P1ND4v2 [40,41,42].

The Wuhan group actually conducted the first phase 1 clinical trial (www.clinicaltrials.gov, accessed on 25 October 2020, NCT01267422) for LHON at the Huazhong University of Science and Technology, Wuhan, China, in 2011 [43,44,45]. They initiated two multicentric non-randomized studies (NCT03153293/NCT03428178) in 2017 and 2018, respectively [46,47]. The Wuhan group evolved to establish Wuhan Neurophth Biotechnology Limited Company in 2016. They started a phase 1/2/3 trial (GOLD, NCT04912843) in 2021, which consists of two parts: part 1 is a dose-escalation study, and part 2 is a randomized, double-blind, sham-injection control study to verify the efficacy and safety of NR082 (rAAV2-ND4).

Due to the rapid development of gene therapy for LHON, the topic warrants a new review to report the existing evidence. Therefore, we conducted a systematic review of gene therapy for LHON.

## 2. Methods

Our study was conducted in accordance with the PRISMA guidelines. Institutional review board approval was waived because of the nature of the study.

### 2.1. Search Strategy and Study Selection

Our systematic review included studies that met the following criteria: (1) the study recruited patients with LHON, and (2) the intervention was gene therapy. We searched PubMed, Cochrane Library, and EMBASE until February 2022 without language limitations, and we checked registered trials at clinicaltrials.gov and the EU Clinical Trials Register. In addition, we also collected some unpublished data from the latest news on the valid official website. Our search strategy was composed of terms for Leber hereditary optic neuropathy and gene therapy, including “Leber hereditary optic neuropathy”, “Leber hereditary optic atrophy”, “Leber optic atrophy”, “LHON”, “gene therapy”, “recombinant gene”, and “gene delivery”. The details of our search strategy are documented in the Appendix A. After removing duplications, two reviewers (SCC and HCC) independently screened titles and abstracts and then retrieved the full texts of potential target papers for further review (Figure 3). We also used a snowballing approach to confirm existing trial protocols and further identified a trial at clinicaltrials.gov, NCT03428178, that could not be found with the keyword “Leber hereditary optic neuropathy”.

### 2.2. Quality Assessment

Methodological bias was assessed independently by 2 reviewers (SCC and HCC). For randomized control trials (RCTs), we used the Cochrane Risk of Bias tool (ROB). Potential bias was assessed in three aspects and with seven items, namely, random sequence generation, allocation concealment, blinding of participants and personnel, blinding of assessment, incomplete outcome data, selective reporting, and other sources of bias. For non-randomized trials, the Newcastle–Ottawa Scale (NOS) tool was applied, and potential bias was evaluated using eight elements within three domains: Selection, Comparability, and Outcome. A third reviewer (AGW) would assess the quality of an article if disagreement between two reviewers occurred.

### 2.3. Data Extraction

Two reviewers (SCC and CHC) extracted data, including the following information: identifier of the clinical trial, study year, location, study design, study population, primary outcome, and secondary outcome, including best-corrected visual acuity (BCVA) improvement and safety profile. We extracted the mean and standard deviation (SD) for continuous outcomes and extracted events and sample size for binary outcomes.

## 3. Results

### 3.1. Patient Characteristics

We included 18 references [37,38,40,41,42,43,44,45,46,47,48,49,50,51,52,53,54,55] from 3 RCTs [37,38,48,49,50,51], 3 non-randomized control trials, and 2 continual observation trials [40,41,42,43,44,45,46,47,52,53,54,55]. In addition, three studies without published articles were included after reviewing clinicaltrials.gov. Information on the registration of these 11 trials and the characteristics of the included trials are listed in Table 1. The included trials with published data comprised a total of 253 patients. The mean age of patients ranged from 19 to 47.9 years old. Most of the patients were male due to the disease’s nature. Most of the trials (n = 6) were sponsored by GenSight Biologics, France. One trial was sponsored by Bascom Palmer Eye Institute, University of Miami, USA. The remaining four trials from China were sponsored by Huazhong University of Science and Technology or Wuhan Neurophth Biotechnology Limited Company.

### 3.2. Risk of Bias Assessment

Generally, the quality of the three included RCTs was high, with few “unclear” and “high risk”. “Some concerns” were raised for the risk of bias in the domain of random sequence generation due to incomplete information in the trial registry and article. “High risk” in the blinding domain was identified for the trial NCT2064569 because it was an unmasked study. The four included observational studies with available data (NCT02161380, NCT03153293, NCT03406104, and EUDRACT N° 2013-001405-90) were assessed and qualified. The scores ranged from 6 to 7, suggesting the moderate to high quality of these trials. The trial NCT01267422 could not be assessed because it was a single-arm study. The details of the risk of bias assessment are listed in the Appendix A.

### 3.3. Trial Design and BCVA Change: Bilateral VA Improvement after Unilateral Injection

Seven trials reported the BCVA change as the visual outcome. Most of the studies reported the BCVA change from baseline. The visual outcomes of three different groups are reviewed in the following.

Wan et al. [52] (Wuhan group) recruited nine patients into their phase 1 study (NCT01267422) in 2011. Among nine patients, eight patients received a single intravitreal injection of rAAV2-ND4 in their poor-vision eye, and one patient received a bilateral injection. After 9 months of follow-up, they found that six of nine patients had VA improvement of at least 0.3 logMAR in both eyes [52]. They further performed a subgroup analysis according to disease duration and found that patients with ≤2 years’ disease duration had a mean BCVA improvement of 0.30 logMAR in the treated eye and 0.35 logMAR in the untreated eye from baseline, whereas patients with >2 years’ disease duration had a mean BCVA improvement of 0.40 logMAR in the treated eye and 0.25 logMAR in the untreated eye at 36 months post-treatment [44]. In 2017, the Wuhan group initiated another study (NCT03153293) [46], which recruited a total of 149 patients who received a single unilateral intravitreal injection of rAAV2-ND4. They found that 54 patients had a rapid significant VA improvement from baseline (more than logMAR 0.3) in at least one eye within 3 days of treatment. They divided patients according to the response time to therapy and tried to identify confounding factors of treatment outcomes. They found that the period between onset and treatment and baseline BCVA was significantly associated with rapid improvement in VA for both the injected and non-injected eyes. However, they could only detect a significant association between age and rapid VA improvement in the injected eye, not in the non-injected eye [46]. Xin et al. [55] conducted a subgroup analysis of 40 enrolled patients. They found that a single intravitreal injection might lead to bilateral VA improvement from baseline (0.21 logMAR in the injected eye; 0.24 logMAR in the uninjected eye) at 12 months post-treatment. The same group initiated another new multicentric non-randomized intervention trial (NCT03428178) in 2018. They included 120 patients with various onset times of the disease receiving a single intravitreal injection of rAAV2-ND4. The data from this trial have not been published yet. Wuhan Neurophth Company initiated a new trial, the GOLD trial (NCT04912843), in 2021 as a randomized, double-blind, sham-injection control study comparing the efficacy and safety of AAV2-ND4 (NR082), but the results are not available yet. 

The Miami group conducted an open-label dose-escalation phase 1 trial (NCT02161380) in 2014. They initially recruited five patients and divided them into three groups, namely, the bilateral chronic group, bilateral acute group, and unilateral acute group. The patients received a single unilateral intravitreal injection of scAAV2-P1ND4v2 at three escalation doses. One patient from the low-dose group and one patient from the medium-dose group had significant VA improvement at 90 days post-treatment. However, there was no significant improvement in the remaining three patients [41]. They further enrolled nine more patients in this study later. They reported a mean VA improvement of 0.24 logMAR in the treated eye and 0.09 logMAR in the fellow eye compared to baseline in the bilateral chronic group and bilateral acute group at 12 months post-treatment. However, there was no significant improvement in the unilateral acute group [40]. 

GenSight Biologics (Paris group) initiated a dose-escalation study in 2014 (NCT02064569), and the 5-year follow-up data were recently published (EUDRACT N° 2013-001405-90). They recruited 15 patients who received a single unilateral intravitreal injection of rAAV2/2-ND4 (GS010) at four escalating doses. According to the article, a durable VA improvement of 0.44 and 0.49 logMAR compared with baseline in the treated and untreated eyes, respectively, was shown at 5 years post-treatment [54]. 

GenSight Biologics further conducted two phase 3 trials, RESCUE (NCT02652767) and REVERSE (NCT02652780), in 2016 [48,51]. These two trials were randomized, double-masked, sham-controlled studies with a single unilateral intravitreal injection of rAAV2/2-ND4 (GS010, LUMEVOQ). The difference in trial design between these two trials was the inclusion criteria: the RESCUE study included participants who had a duration of vision loss of less than 6 months in the first affected eye. On the other hand, the REVERSE study included participants with vision impairment for between 6 and 12 months. The RESCUE and REVERSE trials recruited 38 and 37 patients, respectively. The primary endpoints of these two studies were the difference in the change from baseline in BCVA between the treated eye and sham-treated eye. However, the primary endpoint was not met due to unexpected bilateral VA improvement in both eyes. After 96 weeks of follow-up, the change in BCVA from baseline was 0.18 (logMAR, LS mean) in the treated eye and 0.21 (logMAR, LS mean) in the sham-treated eye in RESCUE and 0.31 (logMAR, LS mean) in the treated eye and 0.26 (logMAR, LS mean) in the sham-treated eye in REVERSE [48,51]. Notably, the baseline VA of patients was much worse in REVERSE. GenSight Biologics published the long-term follow-up data from RESCUE/REVERSE as RESTORE (NCT03406104) in 2021. In this combined cohort, half of the eyes (53.8%) received treatment around 6 months after vision loss, and nearly all eyes (92.7%) were treated within 1 year. The non-parametric locally estimated scatterplot smoothing regression (LOESS) analysis of 152 eyes of subjects in RESCUE and REVERSE showed a progressive and sustained BCVA improvement, with mean BCVA steadily improving from 1.57 logMAR at 12 months after vision loss to 1.26 logMAR at 48 months after vision loss [49]. 

GenSight Biologics conducted another new phase 3 trial, REFLECT (NCT03293524), which began in 2018 and completed enrollment in 2019. They included 98 patients with vision loss within 1 year and divided patients into a bilaterally treated arm and unilaterally treated arm. The preliminary data were released on the GenSight Biologics website [56]. Bilaterally treated subjects had a VA improvement of 0.23 logMAR in the first affected eye and 0.15 logMAR in the second affected eye from baseline, whereas unilaterally treated subjects had a VA improvement of 0.15 logMAR in the first affected (treated) eye and 0.08 logMAR in the second affected (placebo) eye from baseline at 78 weeks. Notably, they reported that the efficacy was more clearly demonstrated in VA improvement from nadir (0.37 logMAR in the first affected eye and 0.31 logMAR in the second affected eye in bilaterally treated subjects; 0.37 logMAR in the first affected eye and 0.25 logMAR in the second affected eye in unilaterally treated subjects). They also reported that bilateral treatment with LUMEVOQ (GS010) was more beneficial than unilateral treatment.

### 3.4. Optical Coherence Tomography (OCT)—Inconsistent Results

Overall, six trials [41,43,46,48,51,54] reported the outcomes of ganglion cell layer thickness or retinal nerve fiber layer thickness. Some of the studies reported no significant change in the anatomical outcomes of OCT [41,43,46,48]. Vignal-Clermont C et al. [54] reported that treated eyes had a mean decrease of 7.92 μm in RNFL after 5 years of follow-up. Decreases in both RNFL thickness and GCL macular volume were also detected in the RESCUE trial [51]. However, the phenomenon could not be found in the REVERSE trial [48], which had a similar trial design to the RESCUE trial. This probably resulted from greater baseline RNFL/GCL thickness among RESCUE patients, who were in the earlier stages of the disease. In addition, Yang et al. [43] reported that the change in the OCT results was not significantly associated with VA improvement.

### 3.5. Humphrey Visual Field (HVF)—Inconsistent Results

Six trials [41,43,46,48,51,54] reported outcomes for the visual field. Most of the studies used Humphrey visual field perimetry to evaluate the visual field. The results were heterogeneous. Yang et al. [43] reported that significant improvements in MD and VFI were noted in the visual field, and the changes in MD and VFI were associated with BCVA improvement. However, Liu et al. [46] reported that the change in MD was not associated with BCVA change. Feuer et al. [41] could only detect a significant difference in VF change between the treated eye and fellow eye, but the difference was caused by the worsening of the fellow eye due to disease progression. Vignal-Clermont C et al. [54] reported that patients had a mean improvement of 2.66 dB in MD after a 5-year follow-up. The results of RESCUE revealed that the mean MD worsened until week 48 but remained stable from week 48 to week 96 in both eyes [51]. On the other hand, the patients in REVERSE had a mean improvement of 2.70 dB in MD in the treated eye and 2.57 dB in MD in the fellow eye at week 96 [48].

### 3.6. Quality of Life (QOL)—Improvement in Several Scales of National Eye Institute Visual Function Questionnaire 25

Two trials, RESCUE and REVERSE, provided available data on QOL [48,51]. They used National Eye Institute Visual Function Questionnaire 25 for evaluation. In the RESCUE trial, increased scores were noted in subscales such as mental health, role difficulties, and dependency, but no significant increases in subscales such as near-vision activities and social functioning. Furthermore, there was no significant difference in the mean Composite score either. However, in the REVERSE trial, patients had a significant increase in the Composite score and most of the subscales, excluding ocular pain.

### 3.7. Safety Profile and Dose—Generally, No Severe Adverse Events Were Noted

A total of seven trials [41,43,46,48,51,54,55,56] reported safety profiles. Generally, there were no serious ocular adverse events or systemic adverse events related to gene therapy. Several minor adverse events were reported. There were three phase 1–2 studies. Yang et al. used doses of 5 × 10^9^ vg for patients younger than 12 years old and 1 × 10^10^ vg for patients older than 12 years old. They reported that there were no ocular or systemic adverse events related to gene therapy in these patients [43]. The Miami group first recruited five patients and determined safety and tolerability in these patients. They tried three doses in their studies: low dose (5 × 10^9^ vg), medium dose (2.46 × 10^10^ vg), and high dose (1 × 10^11^ vg). Generally, no severe ocular or systemic adverse events were noted, and no dose–response for visual improvement was observed in this study [41]. GenSight Biologics conducted a phase 1/2 dose-escalation trial. They enrolled patients in four cohorts with four doses: 9 × 10^9^ vg, 3 × 10^10^ vg, 9 × 10^9^ vg, and 1.8 × 10^11^ vg, and found that the optimal dose was 9 × 10^9^ vg, which was then chosen as the treatment dose for the following clinical trials conducted by GenSight Biologics. In addition, they reported that 90 of the 96 adverse events (94%) were mild in intensity, and the three most common ocular adverse events were anterior chamber inflammation, vitritis, and ocular hypertension. In addition, there were no serious ocular or systemic adverse events related to gene therapy or the treatment procedure. According to the data from RESCUE and REVERSE, the most frequent ocular adverse event was ocular inflammation, which was documented in 74% of eyes in RESCUE and in 92% of eyes in REVERSE [48,51]. Ocular inflammation was limited to anterior and intermediate uveitis. In other words, no posterior uveitis affecting the retina or optic nerve was ever reported. The preliminary data from REFLECT showed a good safety profile in both unilaterally and bilaterally treated patients. There were no severe systemic or ocular adverse events [56].

## 4. Discussion

### 4.1. Key Findings

This systematic review included five RCTs (two without published articles), four non-randomized control trials (one without published articles), and two continual observation trials. There were three phase 1/2 studies, which were conducted by the Wuhan group (Huazhong University of Science and Technology), the Miami group (Bascom Palmer Eye Institute), and the Paris group (GenSight Biologics). The Wuhan group reported that gene therapy was well tolerated, and significant bilateral visual improvement was noted in a certain proportion of unilaterally treated patients. The Miami group did not find a dose–response effect, but a good safety profile was proved, and a VA improvement of more than 0.3 logMAR was noted in some patients. GenSight Biologics initiated a dose-escalation phase 1/2 trial comprising four cohorts with different doses. They found that a dose of 9 × 10^9^ vg had a better benefit/risk ratio. Overall, no severe adverse events related to gene therapy occurred. 

GenSight Biologics conducted three randomized phase 3 trials, RESCUE and REVERSE in 2016 and REFLECT in 2018. In the RESCUE and REVERSE studies, they reported that unilaterally treated patients had bilateral VA improvements from baseline. Notably, the VA change from baseline was greater in REVERSE, which included patients with a longer period of vision loss. The preliminary data from REFLECT revealed a larger treatment effect for bilaterally treated subjects than unilaterally treated subjects and a favorable safety profile in all cases. The outcomes of VF and OCT were heterogeneous. This might have resulted from different inclusion criteria for disease severity and period of vision loss in these studies. Overall, gene therapy might lead to bilateral visual improvement following unilateral intravitreal treatment and was well tolerated. The most common ocular adverse events were mild to moderate ocular inflammation. 

### 4.2. Bilateral Visual Improvement after Unilateral Injection

A contralateral effect of gene therapy following unilateral injection was unexpectedly discovered in these trials. In RESCUE and REVERSE, the primary endpoint, which is defined by a difference of 0.3 logMAR between the treated eye and sham-treated eye, was not met. The researchers tried to identify the mechanism of the contralateral effect of gene therapy. Therefore, a nonhuman primate study enrolled cynomolgus monkeys and treated them with unilateral intravitreal injection [48]. Viral DNA was detected in the contralateral eye and visual pathway in all three animals after 3 months of follow-up. The researchers hypothesized that the viral vector DNA may have transferred to the contralateral eye via the optic nerve and optic chiasm. A similar phenomenon of the transneuronal spread of viral vectors had been reported in a previous study [57]. Brain plasticity may be a possible alternative mechanism of VA improvement in the contralateral eye [58]. Furthermore, transfer of mitochondrial material via interconnected astrocytic processes could be another possible mechanism [59]. 

To better quantify the efficacy of gene therapy, a study comparing the visual outcomes of treated patients with those of untreated patients is warranted. The ongoing study in China (NCT04912843) may give further insight into this specific aim in the future. In 2021, GenSight Biologics conducted a meta-analysis of the indirect comparison of intravitreal gene therapy with natural history in LHON patients [60]. 

### 4.3. Comparing the Intravitreal Gene Therapy vs. Natural History 

Newman et al. [60] enrolled 76 patients from two randomized controlled trials, REVERSE and RESCUE, and their joint extension trial into the intervention group. On the other hand, they enrolled 208 patients from 11 studies without limitations on the trial design into the natural history group. Notably, studies in which patients received treatment with idebenone were not excluded. They strictly performed the statistical method, applying the LOESS model to compare the visual outcomes at 12, 18, 24, 36, and 48 months after vision loss. The mean difference in VA between the treated eye and natural history eye was 0.33 logMAR, and a significant difference in mean VA between treated eyes and natural history eyes was found at all time points. They further performed a multivariate analysis of VA improvement and reported that patients with younger age and a shorter follow-up period had more VA improvement. The review methodology was carried out to a high standard and was sufficiently described in their trial. However, there were still some concerns in this systematic review and meta-analysis. Trials that recruited patients in the natural history cohort were highly heterogeneous regarding trial design and population. 

### 4.4. Future Perspectives and Limitations

Due to the bilateral visual improvement after unilateral intravitreal injection, prior clinical trials for LHON failed to meet the primary endpoint. Thus, a new RCT comparing treated patients with untreated patients is still warranted. Moreover, researchers from the Huazhong University of Science and Technology [43,46] tried to identify important factors for visual prognosis. They found that age and the period of the disease may be confounding factors of VA change. However, there were insufficient data for subgroup analysis on this topic. We hope that a subgroup analysis can be performed in a future trial to clarify the confounding factors of VA outcome. Furthermore, a dose–effect was noted in the ongoing REFLECT study. Bilateral treatment seems to have more favorable visual outcomes than those of unilaterally treated patients. Bilateral injection could potentially become the treatment of choice for LHON in the future. For poorly responsive patients, we presume that supplemental injections may be considered.

Our study has the following limitations. First, the included trials presented heterogeneity regarding the trial design and recruited population. We were unable to further perform a meta-analysis in our study. Second, currently, data from high-quality RCTs are limited. Finally, according to the current evidence, we cannot confirm the clinical efficacy of gene therapy for LHON yet. A randomized clinical trial comparing the gene therapy group with a parallel placebo group would be ideal for this topic.

## 5. Conclusions

We critically appraised existing trials that investigated gene therapy in m.11778G > A LHON patients. There were three phase 1/2 trials from three research groups. These studies indicated the safety of gene therapy for LHON. There were four phase 3 trials from two research groups, with two trials completed with published references and one with preliminary data. Overall, gene therapy seems to induce bilateral VA improvement in LHON. No major adverse events were reported in the included studies. Several trials are still ongoing, and we expect and await their results. Further studies are warranted to clarify the efficacy, confounding factors, and dose–response effect in gene therapy.

## Figures and Tables

**Figure 1 biomedicines-10-01930-f001:**
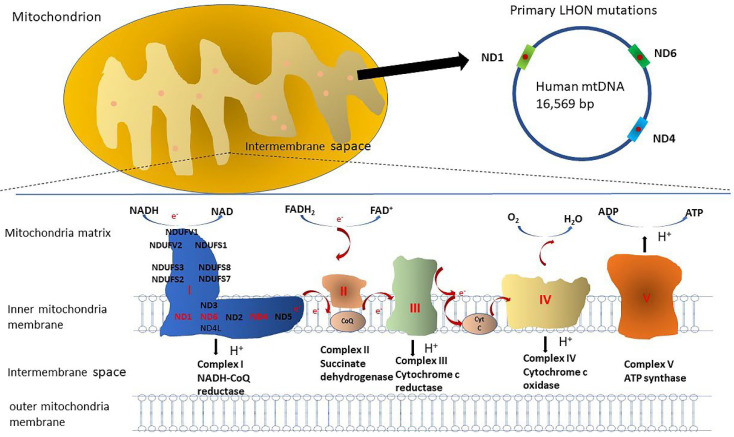
The diagram represents the electron transport chain and the three common primary mutations at mitochondrial nucleotide positions 3460, 11,778, and 14,484. The mutations affect ND1, ND4, and ND6 subunit genes of complex I, respectively.

**Figure 2 biomedicines-10-01930-f002:**
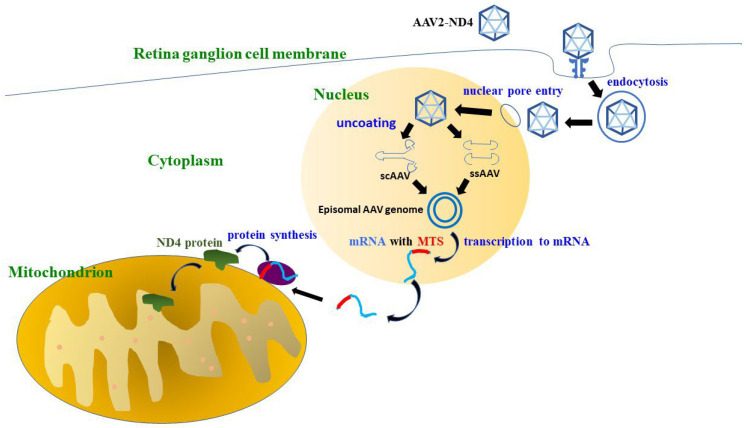
The diagram of the AAV2-ND4 transduction pathway. Clathrin-mediated endocytosis is triggered after AAV2-ND4 is recognized by glycosylated surface receptors on the host cells. Then, AAV2-ND4 passes through the cytosol, shuttles into the nucleus through the nuclear pore, and is uncoated. Single-stranded AAV (ssAAV) and self-complementary AAV (scAAV) transcribe the ND4 mRNA with a mitochondrial targeting sequence (MTS), which helps ND4 mRNA to be delivered to the mitochondrial surface. The ND4 protein is then synthesized and imported into the mitochondrion.

**Figure 3 biomedicines-10-01930-f003:**
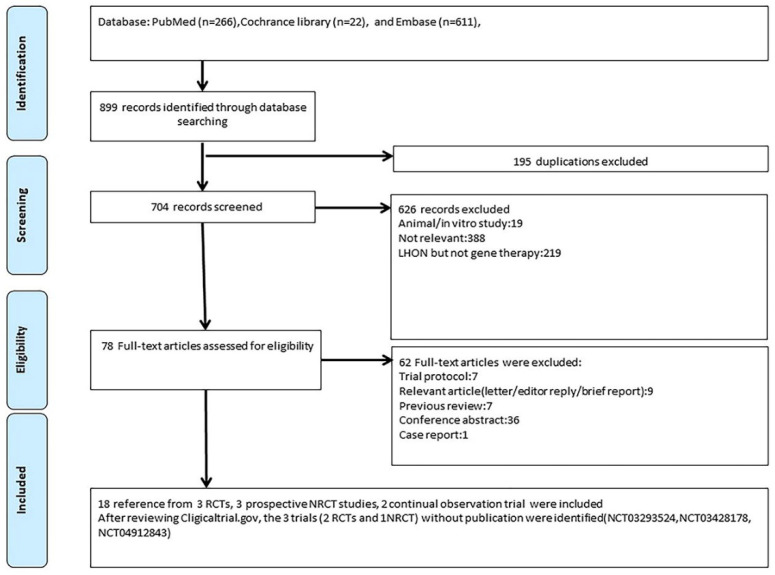
Flowchart of study selection.

**Table 1 biomedicines-10-01930-t001:** Trial registration and characteristics of included trials.

Trial ID	Study Phase	Year	Follow-Up Time	Patients Number	Age (Mean)	Gender (M/F)	Randomized or Not	Trial Design	Location	Outcomes
NCT01267422	1	2011	36 months	9	19.22	7/2	Non-randomized	Dose-escalation Single-arm	Wuhan, China	BCVA//HVF/OCT/safety profile/VEP
NCT02064569	1/2	2014	12 months	15	47.9	13/2	Randomized	Dose-escalation Four arms	Paris, France,	Safety profile
NCT02161380	1	2014	6 months	5	43.00	4/1	Non-randomized	Dose-escalation Three arms Bilateral chronic group Bilateral acute group Unilateral acute group	Miami, Florida,	BCVA/HVF/neutralizing antibodies and quantitative PCR/OCT/pattern ERG/safety profile
NCT02652767 (Rescue)	3	2016	24 months	38	36.8	31/7	Randomized	Two arms: Treated eye Sham-treated eye	United States/France/Germany /Italy/United Kingdom	BCVA/contrast sensitivity/HVF/OCT/QOL/safety profile
NCT02652780 (Reverse)	3	2016	24 months	37	34.2	29/8	Randomized	Two arms: Treated eye Sham-treated eye	United States/France/Germany /Italy/United Kingdom	BCVA/contrast sensitivity/HVF/OCT/QOL/safety profile
NCT03153293	2/3	2017	12 months	149	19	131/18	Non-randomized	Two arms: Rapid response arm Slow response arm	Wuhan, Hubei,	BCVA/HVF/OCT
NCT03293524 (REFLECT)	3	2018	12 months	98	N/A	N/A	Randomized	Two arms: Bilateral treatment arm Unilateral treatment arm	United States/Belgium/France/Italy/Spain/Taiwan/United Kingdom	BCVA/contrast sensitivity/HVF/OCT/QOL/responder analysis/safety profile
NCT03406104	3	2016	60 months	61	35.1	48/13	Follow-up study of RESCUE and REVERSE	Two arms: Treated eye Sham-treated eye	United States/France/Germany /Italy/United Kingdom	BCVA/QOL
NCT03428178	N/A	2018	12 months	120	N/A	N/A	Non-randomized	Five arms according to different periods of disease onset	Wuhan, China	BCVA/HVF/OCT/VEP/liver and kidney function
NCT04912843	1/2/3	2021	13 months	102	N/A	N/A	Randomized	Part one: dose finding Part two: Treatment group Sham injection group	Beijing, China	BCVA/contrast sensitivity/cell immunogenicity/fluids immunogenicity/HVF/QOL safety profile/vector biodistribution
EUDRACT N° 2013-001405-90.	1/2	2014	60 months	15	47.9	13/2	Follow-up study of NCT02064569	Fours arms with a fifth cohort (expansion)	Paris, France	BCVA/contrast sensitivity/HVF/OCT/pattern ERG/QOL/VEP/safety profile

BCVA = best-corrected visual acuity; HVF = Humphrey visual field; OCT = optical coherence tomography; ERG = electroretinogram; QOL = quality of life; VEP = visual evoked potential.

## Data Availability

Data are available on request.

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
