# Peer review of "Leber Hereditary Optic Neuropathy: Molecular Pathophysiology and Updates on Gene Therapy"

_biomedicines, 2022, doi:10.3390/biomedicines10081930_

Round 1

Reviewer 1 Report

I rarely read a manuscript and immediately recommend it for publication.

since the introduction the authors transport the reader progressively in a natural way. certainly the scientific community in addition to deserving the content of the manuscript also deserves the model of exposure and scientific rigor.

personally I wouldn't change one iota, great job.

Author Response

Thanks for your precious comment. We really appreciate for your positive feedback.

Reviewer 2 Report

This manuscript describes a systematic review of gene therapy studies for Leber Hereditary Optic Neuropathy (LHON). The authors found results from several clinical trials of gene therapy for LHON. The methods appear to be described adequately, and the review follows PRISMA guidelines. The analysis is appropriate. The discussion is helpful overall for contextualizing the results. The manuscript would benefit from a couple minor clarifications/changes:

- Figure 1 graphics have low quality; this especially impacts the bottom half of the figure in which some of the text is difficult to read.

- It would be helpful to give a short statement about the mechanism for idebenone (4th page, top) where it is introduced as a treatment for LHON.

- 5th page, paragraph starting “Until now, the most abundant clinical data…”: This paragraph seems a little out of place in the introduction, since it is a description of the clinical trials that will be reviewed later in the manuscript, which are identified in the systematic analysis.

- Along the same lines, the beginning of the discussion (lines 159-195) might fit better in the introduction as it introduces the idea of clinical trials for LHON gene therapy and lays the foundation for why a systematic review is needed. This point and the one prior to it are only suggestions for the authors.

- The figures and table are appropriate and help with understanding the data in this manuscript.

- some acronyms/abbreviations do not appear to have been defined; I recommend double-checking all of these.

- Line 128: “near activates” should be rendered as “near-vision activities.”

Author Response

Thanks for your comment. We tried our best to accomplished the study. We really appreciate for your positive comments and suggestion.

  1. Figure 1 graphics have low quality; this especially impacts the bottom half of the figure in which some of the text is difficult to read.

    Response: Thanks for your kindly comment. We had increased the resolution of the figure and adjusted the font size in the picture.

  2. It would be helpful to give a short statement about the mechanism for idebenone (4thpage, top) where it is introduced as a treatment for LHON.

    Response: Thanks for your comment. We have added the brief statement about the mechanism for idebenone in the part of introducing the treatment of LHON.

  3. 5th page, paragraph starting “Until now, the most abundant clinical data…”: This paragraph seems a little out of place in the introduction, since it is a description of the clinical trials that will be reviewed later in the manuscript, which are identified in the systematic analysis.

    Response : Thanks for your comment. According to the Reviewer’s suggestion, we shortened this paragraph and deleted the redundant content. We try to keep it and make it more concisely, hoping that it can help the reader to get the development timeline of LHON gene therapy. 

  4.  Along the same lines, the beginning of the discussion (lines 159-195) might fit better in the introduction as it introduces the idea of clinical trials for LHON gene therapy and lays the foundation for why a systematic review is needed. This point and the one prior to it are only suggestions for the authors.

    Response : Thanks for your comment. According to the Reviewer’s suggestion, we moved this paragraph (lines 159-195) to introduction, and merged it with the paragraph introducing allotopic expression. The basic principle and the animal study were introduced sequentially. We hope it can help the reader to get the point of the molecular biology of LHON gene therapy.  

  5. The figures and table are appropriate and help with understanding the data in this manuscript.

    Response: Thanks for your comment. We tried to update this important topic and provide the latest information to our reader. We sincerely appreciate for your positive comments.

  6. some acronyms/abbreviations do not appear to have been defined; I recommend double-checking all of these.

    Response: Thanks for your comment. We are sorry for our mistakes. We have defined the acronyms/abbreviations in our article.

  7. Line 128: “near activates” should be rendered as “near-vision activities.”

    Response: Thanks for your comment. We are very sorry for our incorrect writing. We have corrected it